# Alignment of the metatarsal heads affects foot inversion/eversion during tiptoe standing on one leg in demi–pointe position: A cross–sectional study on recreational dancers

Akiko Imura[1]*, Hiroyuki Nagaki[2], Takahiro Higuch[1]

1 Perception and Action Laboratory, Department of Health Promotion Sciences, Tokyo Metropolitan University, Hachioji, Tokyo, Japan, 2 DancingFUN Co., Ltd., Yokohama, Kanagawa, Japan

* akikoimura17@gmail.com

**Data Availability Statement:** This study's minimal underlying data set can be seen in the following:

## Abstract

Classical ballet dancers stand on tiptoe in the demi–pointe position where the ankle is plantarflexed, and the toes extend around a mediolateral axis passing through the second metatarsal head. Foot sickling, the foot inversion/eversion when the forefoot is grounded, should be avoided to achieve esthetics and prevent injuries during tiptoe standing. The foot inversion/eversion angle may change depending on the metatarsal heads through which the toe extension axis passes. This study investigated the relationship between metatarsal alignment in both load positions and foot inversion/eversion angle during tiptoe standing. Nine recreational female ballet dancers performed tiptoe standing on a single leg in the demi–pointe position. The foot inversion/eversion angle, the centre of pressure (COP) positions, and angles between adjacent metatarsal heads in the horizontal plane were investigated using motion–capture data and magnetic resonance imaging of the forefoot. As the angle between the second and adjacent metatarsal heads became more acute during tiptoe standing on the non-dominant leg, the dancers everted the foot more and significantly loaded the first toe–side more, and vice versa ($r = -0.85$ and $-0.82$, respectively). Then, the load positions were distributed on the distal side of the second metatarsal head. These were not seen during standing on the dominant leg with COPs more proximal to the second metatarsal head. In conclusion, dancers load the distal part of the second metatarsal head during tiptoe standing on the non–dominant leg. When the angle at the second metatarsal head was acute, within the triangle formed by the first, second, and third metatarsal heads, even slight mediolateral shifts of load positions altered the toe extension axis around that metatarsal head; the dancers loaded medial to the second metatarsal head and everted the foot and vice versa. Therefore, the angle between the second and adjacent metatarsal heads influenced the foot inversion/eversion angle.

10.6084/m9.figshare.20479893 10.6084/m9.
figshare.21153115.

**Funding:** Tokyo Metropolitan University supported this study. The name of the fund is; Grant-in-Aid for Research on Priority Area. The funders had no role in study design, data collection and analysis, decision to publish, or preparation of the manuscript.

**Competing interests:** The authors have declared that no competing interests exist.

## Introduction

In classical ballet, dancers frequently perform tiptoe standing in the demi–pointe position, which involves toe extension and ankle plantarflexion by approximately 90˚ (Fig 1) [1]. Foot sickling, a combination of ankle inversion/eversion and forefoot abduction/adduction [2], is a concern among dancers. The sickled-out and sickled-in implies foot inversion and eversion, respectively, as the forefoot is almost fixed in tiptoe standing. For ease of understanding, foot sickling is referred to the foot inversion/eversion. The foot inversion/eversion should be avoided during grounding in the demi–pointe position for ballet esthetics [2] and injury prevention. A lateral ankle sprain due to foot inversion with extreme ankle plantarflexion is one of the most typical and crucial injuries [3–6] and results in chronic conditions and secondary complaints [7–9]. Medial ankle sprain due to foot eversion [10, 11] has been less studied but requires longer treatment and recovery periods than a lateral ankle sprain [12, 13]. Dancers must control foot inversion/eversion. If dancers know how the foot inverts/everts in the demi–pointe position, they can predict the foot motions and avoid sudden and severe injuries. Previous studies have reported that the angle and torque of foot inversion are smaller when the inversion is predicted than when it is unpredicted [14, 15]. Dancers may better control foot motions by knowing their tendency for foot inversion/eversion during tiptoe standing in the demi–pointe position.

The geometry of the metatarsal head alignment can influence foot inversion/eversion. The toe extension occurs around an axis that passes through all the metatarsal heads and intersects at an angle ranging from 50–70˚ to the foot longitudinal axis through the second metatarsal head [16]. Not all the metatarsal heads necessarily align in a straight line. Dancers must support their body weight mainly around the second metatarsal head or even at the ball of the first toe during tiptoe standing [2, 17]. In this case, the metatarsal heads comprise the toe extension axis, which may consist of some metatarsal heads rather than all. If the outer three toes are grounded or released, the foot may invert/evert [2]: the toe extension around a vector connecting the first and second metatarsal heads may evert the foot, and that around a vector connecting the second and third metatarsal heads may invert the foot. The angle between the second and its adjacent metatarsal heads (Fig 2A) may influence the vector when load positions scatter around the second metatarsal head. Therefore, the alignment of the metatarsals in the horizontal plane may influence foot inversion/eversion during tiptoe standing in the demi–pointe position.

The above concept is applicable when loading around the second metatarsal head. Depending on which metatarsal head is loaded, the toe extension axis may switch to either vector connecting that metatarsal head and its adjacent metatarsal heads. In the case of standing on both legs, the load position can be changed with a weight shift. Tiptoe standing on a single leg should be investigated to consider the influence of metatarsal alignment. In this case, there is a bias about the leg used to support the whole body in classical ballet dance [19]. Therefore, differences in loading manner should also be considered between both legs. This study aimed to test whether the alignment of the metatarsal heads affects foot inversion/eversion during tiptoe standing on a single leg. We hypothesised that the angles between adjacent metatarsal heads are associated with foot inversion/eversion and the proportion of load on the medial and lateral sides of the foot. In addition, we hypothesised that these relationships are different between both legs.

## Materials and methods

### Experimental setup

Nine female recreational classic ballet dancers participated in this cross–sectional study (age, 47.5 ± 6.59 years; height, 1.60 ± 0.05 m; body mass, 50.9 ± 6.13 kg; ballet experience,

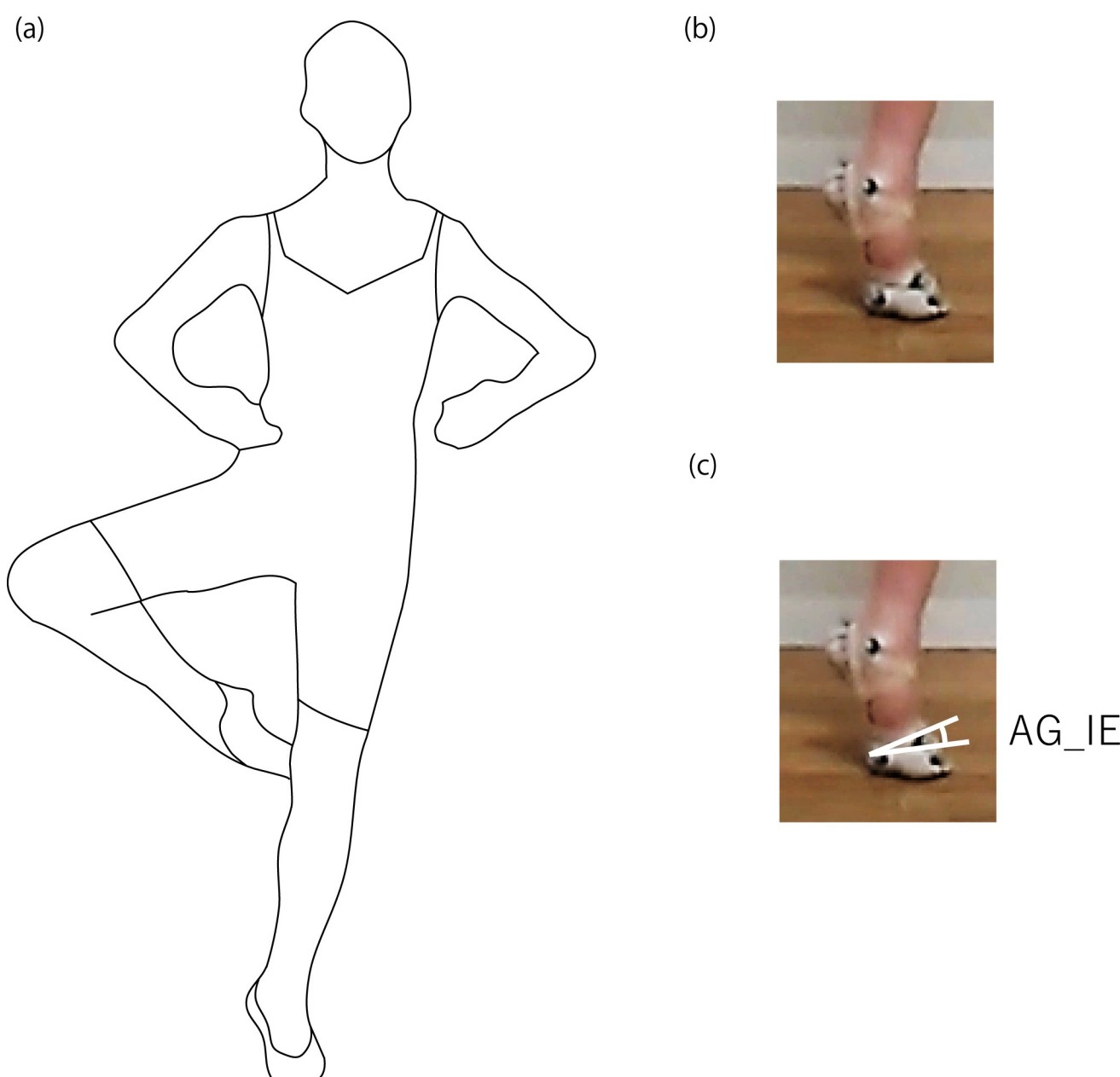

**Fig 1. Posture during tiptoe standing on a single leg.** (a) Dancers kept their balance on a supporting leg without external support. (b) The metatarsophalangeal and ankle joints in the supporting leg are extended and plantarflexed, respectively, during tiptoe standing. (c) The measured foot inversion/eversion angle (AG_IE) in this study.

9.22 ± 2.95 years; training frequency, 3.89 ± 1.83 h per week). They had no disorders in the vestibular system, trunk, or limbs. While some dancers had hallux valgus, a condition in which the tip of the first toe is toward the second toe, none had chronic pain during ballet dancing. For convenience in this study, the leg that each dancer often uses as the supporting leg in ballet dancing is referred to as the non–dominant leg. Conversely, the dominant leg is defined according to the general definition of leg dominance [20]. This study was approved by the ethics committee of Tokyo Metropolitan University, Japan (no.H31-77) and is in accordance with the Declaration of Helsinki. Written informed consent was obtained from all the dancers.

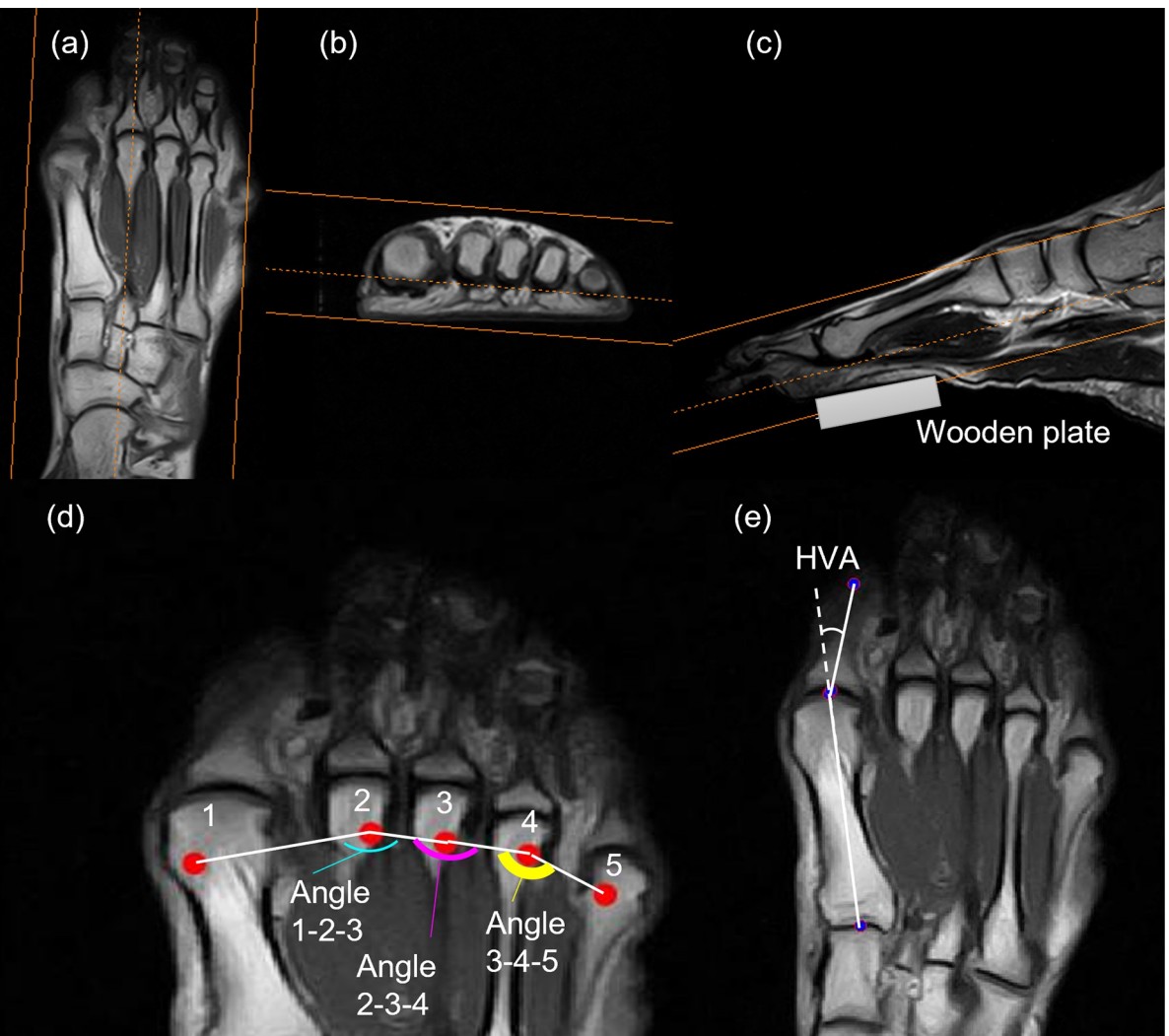

**Fig 2. Magnetic resonance images and definitions of angles representing the alignment of a dancer's metatarsal heads.** (a-c) Views of the horizontal (a), frontal (b), and sagittal (c) images to determine the point on each metatarsal head closest to the sole. (d) Angles between adjacent metatarsal heads. The numbers indicate the first to fifth toes. The thin, medium, and thick arcs indicate angles 1-2-3, 2-3-4, and 3-4-5, respectively. (c) Hallux valgus angle (HVA) determined using the method of Janssen et al. [18].

The experimental task was tiptoe standing on a single leg in the demi–pointe position without any support for as long as possible. After a thorough warm–up, a dancer stood with retro–reflective markers on both legs in an externally rotated position (turnout) and then stood on tiptoe with both hands on the ballet bar, which had a height of 1.0 m. Finally, the dancer removed both hands from the bar while balancing on a single leg. During this task, the toe of the free leg was in contact with the knee joint of the supporting leg, and the upper limbs were placed in front of the chest in a circle (Fig 1). Each dancer performed tiptoe standing while supported on the dominant and non–dominant legs, respectively. Three trials per leg were performed randomly with rest intervals.

Motion capture data were acquired during the trials using standard methods of determinating kinematics and kinetics data in biomechanics [21, 22]. The three–dimensional coordinates of positions of the retroreflective markers attached to the body were obtained using 12 infrared cameras (OQUS 300; Qualisys, Göteborg, Sweden) with sampling frequencies of 250 Hz

through the Qualisys Track Manager Software (Qualisys, Göteborg, Sweden). The positions of the markers on the body landmarks are listed in additional material 1 (S1 File). In addition, a force platform (9286BA; Kistler Inc., Winterthur, Switzerland) with a sampling frequency of 1000 Hz recorded the ground reaction forces (GRFs) and centre of pressure (COP) position of the supporting foot. The cameras and force platform were synchronised electrically.

T1–weighted scans of the forefoot were obtained separately from the tiptoe standing experiments using a magnetic resonance (MR) imaging device (ECHELON Vega; Hitachi, Tokyo, Japan), according to the previous study [23]. A wooden plate (width, 0.12 m; length, 0.05 m; depth, 0.01 m) was tightly fixed to the bottom of the forefoot to prevent the non–weight bearing forefoot from curling (S1 Fig). Then, the forefoot was scanned along the three axes: an axis perpendicular to the line connecting the bottoms of the first and fifth metatarsal heads when viewed from the toe, the longitudinal axis of the second metatarsal bone, and the normal axis to both axes, providing horizontal, frontal, and sagittal images, respectively (Fig 2A–2C). The scan settings were as follows (in order of sagittal, frontal, and horizontal images): time of repetition, 3500, 464, and 380 ms, respectively; time of echo, 100, 26, and 26 ms, respectively; field of view, 210, 180, and 210 $mm^2$, respectively; matrix, 512 pixels for all; slice thickness, 3.6 mm for all; scan duration, 206, 318, and 381 s, respectively.

## Data analysis

**Kinematics and kinetics during tiptoe standing.** Kinematic parameters were processed to determine the joint kinetics during the balance holding phase, according to the previous study [21]. The balance holding phase was defined as the period when the dancer stood on tiptoe with the toe of the free leg in contact with the knee joint of the supporting leg without any support. The data recorded during the longest phase in each trial were analysed and averaged. Marker coordinates and GRF data were processed using fourth–order 0–phase–lag Butterworth low–pass filters with 6 and 150 Hz cut–off frequencies, respectively [24]. The markers on the mediolateral or anteroposterior sides of the joints were used to determine the midpoint as a joint centre (S1 File).

The foot anatomical motions are difficult to present using Euler angles accurately because the anatomical axes defined in the ankle joint and forefoot [25] are not necessarily orthogonal coordinate systems. Therefore, the foot inversion/eversion angle (AG_IE) was determined as the inclination of the vector connecting the markers on the lateral side of the first and fifth metatarsal heads (MP1–MP5 vector) against the floor (Fig 1C). The ankle plantarflexion angle was determined using the longitudinal axes of the foot and lower leg. These angles during the balance holding phase were determined by subtracting the initial angles in the double stance before tiptoe standing. The foot turnout angle was determined as the angle between the longitudinal axes of both feet in the double stance. The hip joint centre was calculated using the functional method [26, 27]. The hip joint and pelvic angles in the global coordinate system were determined as the Euler angle using the segment coordinate systems defined in Table 1. The rotation orders were extension/flexion-abduction/adduction-external/internal rotation and rightward/leftward tilt-forward/backward tilt-leftward/rightward rotation, respectively.

The positions of the COPs were evaluated by projecting them onto the forefoot coordinate system. The forefoot coordinate system was defined using the MP1–MP5 vector (x–axis) and the foot longitudinal axis (y–axis). The medially loaded time (MT) was defined as the time when the COP was on the medial side (MP1 side) to the y–axis, and conversely, the laterally loaded time was on the MP5 side. The MT was normalised with the balance holding phase time.

The ankle joint torques were calculated using the standard inverse dynamics approach with the body parameters of adult Japanese female athlete populations [28] (for details on

**Table 1. Definition of the local coordinate system of the pelvis, thigh, and foot.**

| Segment | Vector | Axes of segment coordinate system |
|---|---|---|
| **Pelvis** | Cross product of the axes of both the unit vector connecting the marker on the midpoint between the right and left posterior superior iliac spines and the unit vector connecting the markers on the left and right anterior superior iliac spines | Longitudinal axis |
| | A unit vector connecting the marker on the midpoint between the right and left posterior superior iliac spines | Mediolateral axis |
| | Cross product of the longitudinal and mediolateral axes | Anteroposterior axis |
| **Thigh** | A unit vector from the knee joint centre to the hip joint centre | Longitudinal axis |
| | A unit vector from the marker on the midpoint of the medial epicondyles of the femur and the tibia to the marker on the midpoint of the lateral epicondyles | Mediolateral axis |
| | Cross product of the longitudinal and mediolateral axes | Anteroposterior axis |
| **Foot** | A unit vector from the marker on the second metatarsal head to the marker on the processus calcaneus | Longitudinal axis |
| | A unit vector from the marker on the inner malleolus to the marker on the outer malleolus | Mediolateral axis |
| | Cross product of the longitudinal and mediolateral axes | Anteroposterior axis |

calculation methods, see the previous study by Winter [24]). Then, the torques were projected onto the axes of the ankle joint coordinate system, which is the same as the foot coordinate system (Table 1). These calculations were programmed using MATLAB software (MathWorks, Natick, MA, USA).

**Measuring angles between metatarsal heads on MR images.** The alignment of the metatarsal head and the hallux valgus angle (HVA) on the scanned MR images were evaluated using MATLAB. The point on each metatarsal head closest to the sole was assumed to represent a loading point during tiptoe standing. An image in which the metatarsal head was identified first when viewed from the bottom was imported to MATLAB. Then, the two–dimensional coordinate of the brightest point of the metatarsal head was determined as the ground point of the bone (see S2 File for the details on determining the point). The angles between the vectors connecting the second and other metatarsal heads were calculated from the coordinates; their abbreviations were angles 1-2-3, 2-3-4, and 3-4-5, respectively (Fig 2D). Hereafter, this study used these angles together in a metatarsal alignment. The HVA was also determined with the method of Janssen et al. [18] (Fig 2E) using MR images, which identified both the metatarsal and proximal phalanges. The images were binarised using adaptive thresholding, and the white and black areas were assumed to be the epiphysis and joint cavity, respectively. The longitudinal axis of the bone was determined by connecting the midpoint of their boundaries. The angle between both longitudinal axes was then calculated as HVA.

## Statistical analysis

The intraclass correlation coefficient (ICC) was used to measure the metatarsal alignment. The correlations between angles representing metatarsal alignment and the mean foot inversion/eversion and MT were assessed using Pearson product–moment or Spearman's rank correlation analyses after testing for normality of distribution using the Shapiro–Wilk test. In addition to these variables, the mean kinematics and kinetics during the balance holding phase were compared using paired–samples Student $t$–tests or Wilcoxon rank–sum tests; the angles representing the metatarsal alignment, balance holding time, COP positions to the second metatarsal head in the forefoot coordinate system, ankle plantarflexion angle, foot turnout

**Table 2. Interclass correlation coefficient for measuring metatarsal alignment.**

| | Intraclass correlation coefficient (1, 3) | 95% Confidence interval | | P value |
|---|---|---|---|---|
| | | Upper | Lower | |
| **Dominant foot** | | | | |
| Angle 1-2-3 | 0.81 | 0.44 | 0.95 | .00 |
| Angle 2-3-4 | 0.75 | 0.25 | 0.94 | .01 |
| Angle 3-4-5 | 0.94 | 0.83 | 0.99 | .00 |
| **Non-dominant foot** | | | | |
| Angle 1-2-3 | 0.95 | 0.85 | 0.99 | .00 |
| Angle 2-3-4 | 0.74 | 0.19 | 0.94 | .01 |
| Angle 3-4-5 | 0.96 | 0.87 | 0.99 | .00 |

angle, ankle joint torques, hip joint angles, and the pelvic angles in the global coordinate system were compared between both legs. Because the rightward/leftward tilt and the leftward/rightward rotation of the pelvis occur symmetrically in both supporting legs, they were compared using their absolute values. The significance level was set at $P < .05$. The false discovery rate (FDR) was adopted for multiple–comparison procedures, with FDR < 0.05 [29]. The statistical powers were calculated in the correlation analyses and comparisons between both legs. These were calculated using MATLAB.

## Results

The ICCs for measuring metatarsal alignments were greater than 0.74 (Table 2). Six dancers had mild or moderately low HVA (that is, >16˚ but <29˚); other dancers had no HVA (Table 3). No significant differences between both legs were identified in the HVA and metatarsal alignment (Table 3).

In the non–dominant leg, significantly negative correlations (r = -0.85) were identified between angle 1-2-3 and both the MT and foot inversion/eversion angle (Fig 3A and 3B). The dominant leg had no significant correlation with the metatarsal alignment (Fig 4).

No significant differences between both legs were identified in the HVA, metatarsal alignment, balance holding time, or angles of the ankle joint and foot (Table 4). The mean COP position was significantly more medial and proximal to the second metatarsal head in the dominant leg than in the non–dominant leg (P = 0.00) (Table 4). The dancers maintained the COP in the medial of the second metatarsal head longer with the dominant leg than with the non–dominant leg (Table 4). The ankle plantarflexion torque was significantly larger in the non–dominant leg than in the dominant leg (P = 0.00) (Table 4).

## Discussion

This study investigated whether the metatarsal alignment affects foot inversion/eversion during tiptoe standing on a single leg in the demi–pointe position. The hypotheses were tested to

**Table 3. Comparison of the alignment of metatarsal bone heads between both legs.**

| | Dominant | Non-dominant | P value | Statistical power |
|---|---|---|---|---|
| **Hallux valgus angle, degrees** | 14.7 ± 4.6 | 16.4 ± 7.2 | .34 | .12 |
| **Angle 1-2-3, degrees** | 148.7 ± 14.7 | 150.4 ± 12.1 | .77 | .06 |
| **Angle 2-3-4, degrees** | 174.7 ± 14.3 | 168.1 ± 9.2 | .05 | .66 |
| **Angle 3-4-5, degrees** | 160.8 ± 6.0 | 164.5 ± 13.3 | .39 | .23 |

*Significantly different between both legs ($P < .05$).

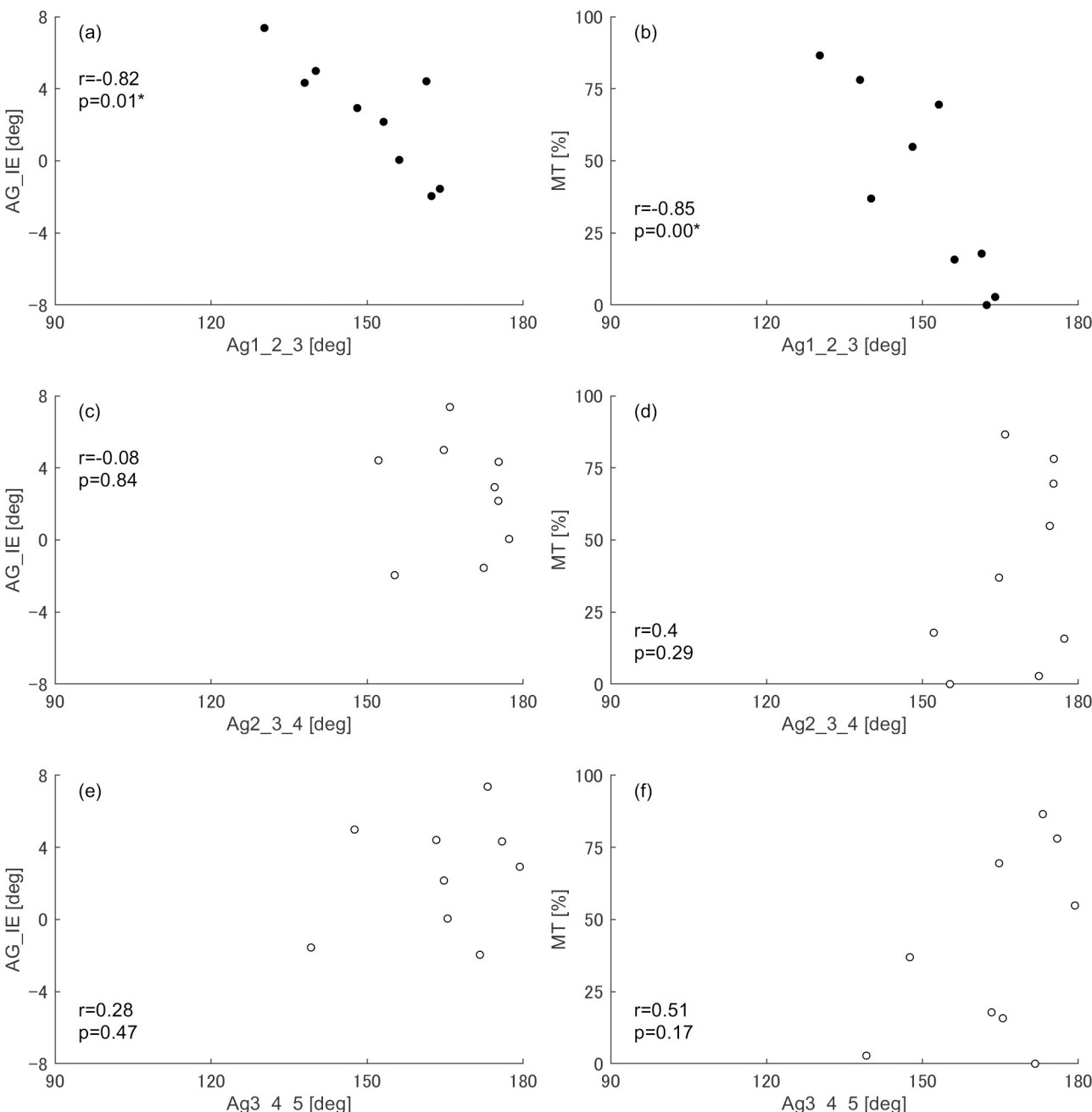

**Fig 3. Correlation between angles representing metatarsal alignment and relevant variables when standing on the non–dominant leg.** AG_IE and MT indicate the mean foot inversion (−)/eversion (+) angle and the % duration when the centre of pressure positions was located medial to the second metatarsal head, respectively, against the total duration of the balancing phase. Graphs with black circle plots indicate significant correlations with the angle representing metatarsal alignment and variables after consideringeffects of the multiple testing. (a) the foot inverts as the angle 1-2-3 becomes significantly wider; (b)% duration to load medially decreases as the angle 1-2-3 becomes significantly wider; (c) the foot inversion/eversion angle does not correlate with the angle 2-3-4; (d) % duration to load medially does not correlate with the angle 2-3-4; (e) the foot inversion/eversion angle does not correlate with the angle 3-4-5; (f) % duration to load medially does not correlate with the angle 3-4-5.

prove the mechanism of metatarsal alignment, whether the load is applied to the second metatarsal head medially or laterally, influencing the foot inversion/eversion angle. The results on the non–dominant leg in this study suggested part of our hypotheses. The foot was likely to evert as angle 1-2-3 became narrower (Fig 3A). Furthermore, as angle 1-2-3 became narrower,

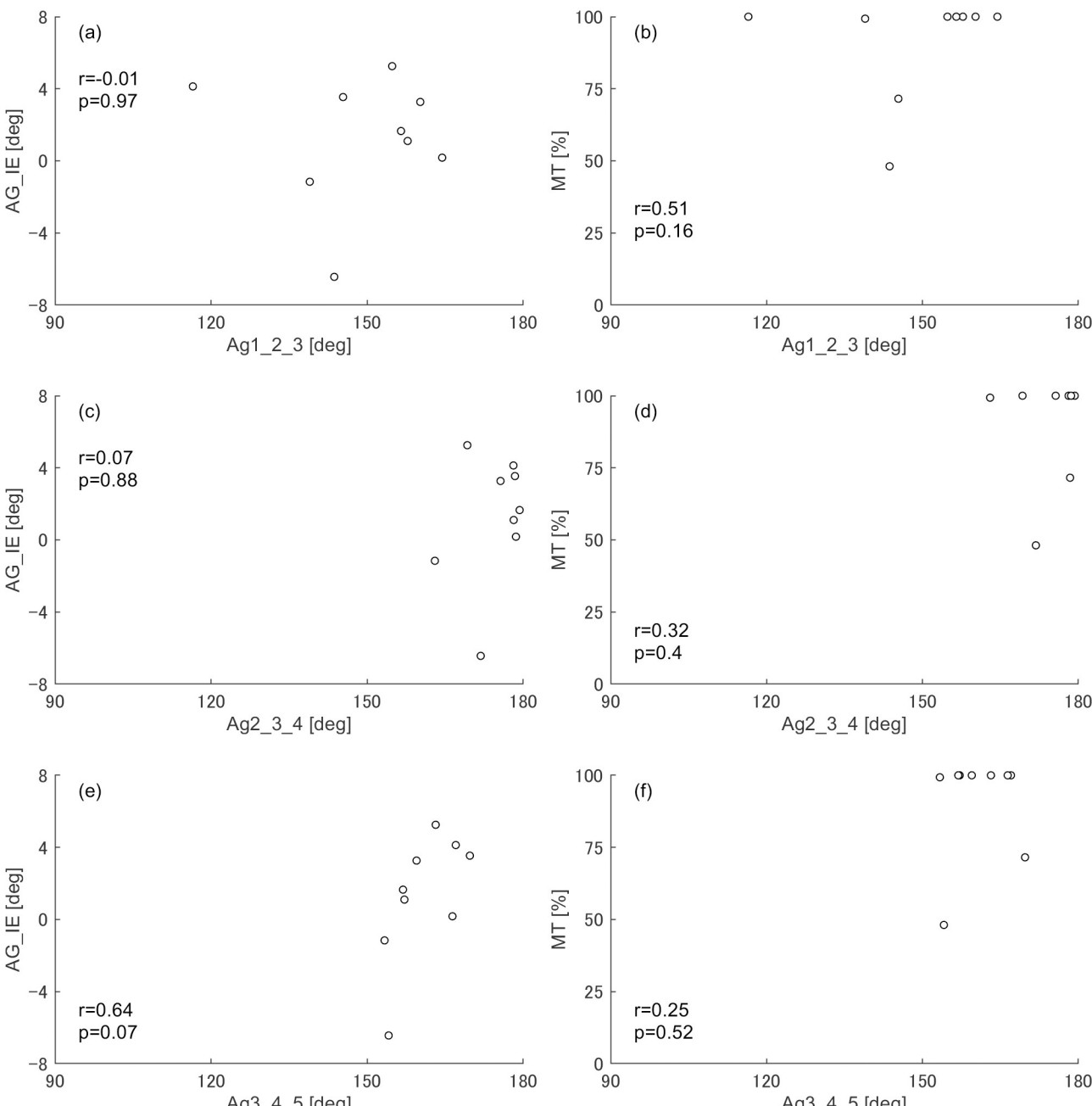

**Fig 4. Correlation between angles representing metatarsal alignment and relevant variables when standing on the dominant leg.** AG_IE and MT indicate the mean foot inversion (−)/eversion (+) angle and the % duration when the centre of pressure positions was located medial to the second metatarsal head, respectively, against the total duration of the balancing phase. Graphs with black circle plots indicate significant correlations with the angle representing metatarsal alignment and variables after considering effects of the multiple testing. (a) the foot inversion/eversion angle does not correlate with the angle 1-2-3; (b)% duration to load medially does not correlate with the angle 1-2-3; (c) the foot inversion/eversion angle does not correlate with the angle 2-3-4; (d) % duration to load medially does not correlate with the angle 2-3-4; (e) the foot inversion/eversion angle does not correlate with the angle 3-4-5; (f) % duration to load medially does not correlate with the angle 3-4-5.

the COP was maintained closer to the MP1 side (Fig 3B). These results indicate that the foot becomes more everted as angle 1-2-3 becomes narrower due to loading on the first toe. Conversely, slight foot inversion and lesser MT were observed in dancers whose angle 1-2-3 was

**Table 4. Comparison of kinematics and kinetics between both legs during tiptoe standing.**

| | Dominant | Non-dominant | P value | Statistical power |
|---|---|---|---|---|
| Balance holding time, s | 4.9 ± 2.7 | 4.0 ± 2.8 | .19 | .11 |
| Dis_MP2-COP$_{ml}$, mm/balance holding time | −12.4 ± 6.9 | 2.6 ± 6.9 | .0004* | .99 |
| Dis_MP2-COP$_{ap}$, mm/balance holding time | −4.6 ± 6.0 | 10.1 ± 3.6 | .0001* | 1 |
| MT, % duration | 91.0 ± 18.6 | 40.3 ± 33.1 | .0009* | 1 |
| Ankle joint (foot) kinematics and kinetics | | | | |
| Plantarflexion, degrees | −34.8 ± 4.4 | −33.3 ± 8.1 | .67 | .1 |
| Inversion/eversion, degrees | 1.3 ± 3.6 | 2.5 ± 3.2 | .082 | .11 |
| Turnout, degrees | 108.6 ± 7.0 | 110.9 ± 4.6 | .30 | .10 |
| Plantarflexion torque, Nm/(kg·m)×10$^{−3}$ | 0.5 ± 0.6 | 1.9 ± 0.4 | .0002* | .99 |
| Pronation/supination torque, Nm/(kg·m) ×10$^{−3}$ | −0.8 ± 0.3 | −0.6 ± 0.4 | .27 | .26 |
| Adduction/abduction torque, Nm/(kg·m) ×10$^{−3}$ | 0.2 ± 0.3 | 0.2 ± 0.2 | .34 | .078 |
| Hip joint angle | | | | |
| Extension, degrees | 5.2 ± 5.7 | 5.9 ± 9.3 | .61 | .053 |
| Adduction, degrees | 0.1 ± 3.1 | 1.1 ± 2.5 | .48 | .096 |
| External rotation, degrees | 18.8 ± 11.0 | 17.3 ± 7.4 | .78 | .059 |
| Pelvic angle in the global coordinate system | | | | |
| Rightward (+) or leftward (−) tilt, degrees | 9.9 ± 3.5 | −10.4 ± 3.3 | .70 | .056 |
| Backward tilt, degrees | 1.9 ± 6.6 | 3.6 ± 7.7 | .021 | .08 |
| Leftward (+) or Rightward (−) rotation, degrees | −10.7 ± 10.5 | 12.2 ± 7.6 | .47 | .059 |

*Significantly different between both legs ($P < .05$). Dis_MP2-COP$_{ml}$/COP$_{ap}$, normalised distance from the marker of the second metatarsal head to the position of the centre of pressure in the mediolateral and anteroposterior directions of the forefoot coordinate system throughout the balance holding phase.

wider (Fig 3A and 3B), indicating that such dancers tended to invert their feet due to loading on the lateral side of the forefoot during tiptoe standing.

The MT changes with angle 1-2-3 during tiptoe standing on the non–dominant leg because the COP positions are strongly influenced by the ankle joint torque [30], and the ankle pronation/supination and abduction/adduction torques may be large when the COP moves to the mediolateral direction from the longitudinal axis of the foot. While the dancers were more loaded on the MP1 side than on the MP5 side when standing on the dominant leg compared with the non–dominant leg, the average magnitudes of the ankle pronation/supination and abduction/adduction torques were not different between both legs (Table 3). However, the ankle plantarflexion torque exerted was greater, and the COP was maintained more on the toe side of the non–dominant leg than that of the dominant leg (Table 3). This suggests that the dancers were loaded around the apex at the second metatarsal head in the triangle formed by the first, second, and third metatarsal heads by exerting a large ankle plantarflexion torque when standing on the non–dominant leg. Therefore, the toe extension axis would switch between a vector connecting the first and second metatarsal heads and that connecting the second and third metatarsal heads from even minor changes in mediolateral loading. The axis connecting the other metatarsal heads may also function as a toe extension axis. However, there was no significant relationship between other angles and the percentage of lateral loading (100% MT because dancers were rarely loaded on the lateral three toes. When standing on the dominant leg, the smaller ankle plantarflexion torque resulted in loading the proximal side to the second metatarsal head, where both axes rarely switch. Therefore, the foot inversion/eversion angle was not influenced by angle 1-2-3 when standing on the dominant leg.

The difference in plantarflexion torque between both legs may be due to different strategies for balancing on tiptoe, as no significant differences in foot geometry were found. The mean

plantarflexion torques exerted during the balance holding phase were much smaller than the maximum voluntary contraction torque [31]. This indicates that the dancers might have adjusted the magnitude of the plantarflexion torque. Furthermore, while the maximum voluntary ankle plantarflexion torque is greater in the dominant leg than in the non–dominant leg in active middle–aged people [32], the plantarflexion torque in this study was greater in the non–dominant leg than in the dominant leg (Table 3). Considering the above information, dancers may strategically change the magnitude of the torque between both legs during tiptoe standing. This may be due to difficulty in leg–trunk joint coordination when maintaining balance. The difference between both legs during tiptoe standing was observed in the pelvic angle in the global coordinate systems; the pelvis tended to lean more backwards when standing on the non–dominant leg than on the dominant leg (Table 3). Dancers would have to rotate the pelvis backwards to balance the forward lean of the body when loading on the toe side. Even in the standing posture, leg and trunk coordination is considered challenging [33–36]. The smaller ankle plantarflexion torque may have been exerted to avoid such difficult leg–trunk coordination during tiptoe standing on the dominant leg.

## Limitations

This study has some limitations. First, the number of participating dancers was small. Effect sizes were large for results in which statistical significance was confirmed; however, some effect sizes were small, especially for kinematic variables in the legs. There might be the risk of a type–two error. However, the p-values and effect sizes of the correlations between the angle 1-2-3 and both AG_IE and MT were significant, supporting a part of the hypotheses. Furthermore, more dancers should be recruited to clarify the mechanics of this relationship in terms of the whole body kinematics and kinetics. Post–hoc analyses for the sample size showed that more than 300 dancers would be required for this study to achieve a medium effect size in comparing both legs. Second, the results of this study may not be directly applicable to particular categories of participants, such as those with differing skill levels or sex. These points may have prevented a clear demonstration of the differences in the anteroposterior rotation angle of the pelvis between both legs. Finally, MR imaging is not an easy method to inform dancers of the metatarsal alignment of their own feet, particularly for dancers who have difficulty accessing medical support. To overcome these limitations, further studies that investigate various categories of dancers using more general methods to measure the metatarsal alignment are required. Despite these limitations, knowing the effect of angle 1-2-3 on foot inversion/eversion when loading the distal side of the second metatarsal head should help dancers prepare for lateral/medial ankle sprains during ballet dancing.

## Conclusion

During tiptoe standing on the non–dominant leg, recreational dancers load the distal part of the second metatarsal head. When the angle at the second metatarsal head of the triangle formed by the first, second, and third metatarsal heads was acute, even slight mediolateral shifts of load positions altered the toe extension axis around that metatarsal head; the dancers loaded medial to the second metatarsal head and everted the foot and vice versa. Therefore, the angle between the second and adjacent metatarsal heads influenced the foot inversion/eversion angle.

## Supporting information

**S1 File. Marker setting.** Positions of reflective markers on the body and anatomical labels are listed.
(DOCX)

**S2 File. Measurement of the angle between adjacent metatarsal heads.** Measurement of the the angle between adjacent metatarsal heads using the MR images are explained in this document.
(DOCX)

**S1 Fig. Wooden plate used in MR imaging.** The plate was fixed on the forefoot using a surgical tape.
(DOCX)

## Acknowledgments

The authors thank Dr Yasuyoshi Mase, the director of HACHIOJI sports Orthopaedic Clinics, for renting out the MRI equipment, Mr Tsunenaga who is a technician of the clinic, and reception staff in the clinic. We would like to thank Editage (www.editage.com) for English language editing.

## Author Contributions

**Conceptualization:** Akiko Imura, Hiroyuki Nagaki.

**Data curation:** Akiko Imura, Takahiro Higuch.

**Formal analysis:** Akiko Imura.

**Funding acquisition:** Akiko Imura.

**Investigation:** Akiko Imura.

**Methodology:** Akiko Imura.

**Project administration:** Akiko Imura.

**Resources:** Akiko Imura, Hiroyuki Nagaki, Takahiro Higuch.

**Software:** Akiko Imura.

**Supervision:** Akiko Imura, Takahiro Higuch.

**Validation:** Akiko Imura.

**Visualization:** Akiko Imura.

**Writing – original draft:** Akiko Imura, Takahiro Higuch.

**Writing – review & editing:** Akiko Imura, Takahiro Higuch.

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
