## [Decision Letter · Decision Letter 0]

13 Sep 2022

PONE-D-22-22601Alignment of the metatarsal heads affects foot inversion/eversion while standing on one leg in a demi-pointe position: A cross-sectional study in dancersPLOS ONE

Dear Dr. Imura,

Thank you for submitting your manuscript to PLOS ONE. After careful consideration, we feel that it has merit but does not fully meet PLOS ONE’s publication criteria as it currently stands. Therefore, we invite you to submit a revised version of the manuscript that addresses the points raised during the review process.

We look forward to receiving your revised manuscript.

Kind regards,

Shazlin Shaharudin

Academic Editor

PLOS ONE

Journal Requirements:

"No"

Reviewers' comments:

Reviewer's Responses to Questions

**Comments to the Author**

1. Is the manuscript technically sound, and do the data support the conclusions?

Reviewer #1: Yes

2. Has the statistical analysis been performed appropriately and rigorously? 

Reviewer #1: No

3. Have the authors made all data underlying the findings in their manuscript fully available?

Reviewer #1: Yes

4. Is the manuscript presented in an intelligible fashion and written in standard English?

Reviewer #1: Yes

5. Review Comments to the Author

Reviewer #1: Summary

The research project investigate the relationship between metatarsal alignment in both load positions and foot inversion/eversion angle during tiptoe standing. Authors hypothesized that the angles between adjacent metatarsal heads are related to foot inversion/eversion and to the proportion of load on the medial and lateral of the foot; Authors also hypothesized these relationships are different between both legs. 9 ballet dancers are collected with experimental design.

Major issues.

Some of figures need clarification. Figure 1 could be improved to emphasize the toe extension and ankle plantarflexion. Does Figure 2 represent the average dancer's metatarsal heads? Where the image source came form? As this image provides a great platform explaining research questions, clear source of images might be needed. It is also not clear how information derived from the results in Figure 3. I suggest the authors need to provide clear explanation in each box.

Minor issues

The authors should check whether there is a reporting guideline suitable for their study; usually, there will be and re-write the manuscript following a guideline checklist. It offers a standard way for authors to prepare report of trial findings, facilitating their complete and transparent reporting, and aiding their critical appraisal and interpretation.

The main limitation is the 9 participants as authors explained in the limitation section, still it should be justified in the text why it is "okay" to conclude your hypotheses with small number of subjects.

The methodology is not clear enough. For example, I don't see any information about motion marker-data collection part. This is a fairly significant omission (or am I missing some part?). Methods to Data analysis should be linked, but this does not seem to be reported in the manuscript.

6. PLOS authors have the option to publish the peer review history of their article (what does this mean?). If published, this will include your full peer review and any attached files.

Reviewer #1: No

---

## [Author Response · Author response to Decision Letter 0]

3 Oct 2022

Reviewer #1: Summary

The research project investigate the relationship between metatarsal alignment in both load positions and foot inversion/eversion angle during tiptoe standing. Authors hypothesized that the angles between adjacent metatarsal heads are related to foot inversion/eversion and to the proportion of load on the medial and lateral of the foot; Authors also hypothesized these relationships are different between both legs. 9 ballet dancers are collected with experimental design.

** 

We would like to express our appreciation to the reviewer for clear understanding of our

study and the helpful suggestions for the improvement of our manuscript. We agree with your comments and have modified our manuscript have revised our manuscript in line with your comments. 

We reconsidered adding explanations about foot inversion/eversion. This study aimed to examine why the supporting foot might turn inward or outward when standing on tiptoe with a single leg. While this foot rotation is called foot sickling–out or sickling–in in dance, non–dance people may be unfamiliar with this word. Therefore, we named it foot inversion/eversion in the manuscript. We mentioned this in the revised manuscript in Lines 53-57.

Major issues.

Some of figures need clarification. Figure 1 could be improved to emphasize the toe extension and ankle plantarflexion. 

** 

We added an enlarged photo of the foot during tiptoe standing, as shown in Fig 1b. Also, we added the angle of AG_IE, the variable presenting the foot inversion/eversion, in Fig 1c.

Does Figure 2 represent the average dancer's metatarsal heads? Where the image source came form? As this image provides a great platform explaining research questions, clear source of images might be needed. 

** 

Figure 2 represents MR images of a dancer in this study. 

It is also not clear how information derived from the results in Figure 3. I suggest the authors need to provide clear explanation in each box.

** 

To find the results of Figures 3 and 4 quickly, we arranged the contents of the Results section, the tables, and the figures (Lines 231-288) and added simple explanations to each box in the caption (Figs 3 and 4). 

To track how to determine these variables quickly, we added the abbreviations used in Figures 3 and 4 into sentences which explain how to determine the variables (Lines 170 & 186). Furthermore, we divided the Data analysis section in the Method into two sub–sections, "Kinematics and kinetics during tiptoe standing" and "Measuring angles between metatarsal heads on MR images".

Minor issues

The authors should check whether there is a reporting guideline suitable for their study; usually, there will be and re-write the manuscript following a guideline checklist. It offers a standard way for authors to prepare report of trial findings, facilitating their complete and transparent reporting, and aiding their critical appraisal and interpretation.

** 

We referred to the guideline for analysing kinematics and kinetics by [21] Derrick et al. (2020) and [22] Leardini et al. (2021). We also referred to [23] Weishaupt et al. (2002), as they measured the distance between the bottom of the metatarsal head bone and the sole using MR images. 

The main limitation is the 9 participants as authors explained in the limitation section, still it should be justified in the text why it is "okay" to conclude your hypotheses with small number of subjects.

** 

The normality of distributions and a type two error are concerns when the sample number is small. We tested the variables after verifying the normality of distribution, and then parametric or non–parametric tests were used accordingly to obtain the accurate p–values. However, the type two error might still occur. Therefore, we added sentences about the type two error caused by the small number of participants, especially in interpreting the limitation of leg kinematics and kinetics results (Lines 345-348). Furthermore, we limited the interpretation of the results to recreational dancers in the title and Conclusion (Line 367).

The methodology is not clear enough. For example, I don't see any information about motion marker-data collection part. This is a fairly significant omission (or am I missing some part?). Methods to Data analysis should be linked, but this does not seem to be reported in the manuscript.

** 

We have added a more detailed explanation of motion capturing according to the guideline for analysing kinematics and kinetics (Lines 133-142). The additional material (S1 File) provides the information on the marker set.

---

## [Editor Report · Decision Letter 1]

5 Oct 2022

Alignment of the metatarsal heads affects foot inversion/eversion during tiptoe standing on one leg in demi–pointe position: A cross–sectional study on recreational dancers

PONE-D-22-22601R1

Dear Dr. Imura,

We’re pleased to inform you that your manuscript has been judged scientifically suitable for publication and will be formally accepted for publication once it meets all outstanding technical requirements.

Kind regards,

Shazlin Shaharudin

Academic Editor

PLOS ONE
---

## [Editor Report · Acceptance letter]

10 Oct 2022

PONE-D-22-22601R1 

Alignment of the metatarsal heads affects foot inversion/eversion during tiptoe standing on one leg in demi–pointe position: A cross–sectional study on recreational dancers 

Dear Dr. Imura:

I'm pleased to inform you that your manuscript has been deemed suitable for publication in PLOS ONE. Congratulations! Your manuscript is now with our production department. 

Kind regards, 

on behalf of

Dr. Shazlin Shaharudin 

Academic Editor

PLOS ONE